# Impact of KRAS Mutation Subtypes and Co-Occurring Mutations on Response and Outcome in Advanced NSCLC Patients following First-Line Treatment

**DOI:** 10.3390/jcm11144003

**Published:** 2022-07-11

**Authors:** Yingjia Sun, Ziming Li, Hong Jian, Liliang Xia, Shun Lu

**Affiliations:** Shanghai Lung Cancer Center, Shanghai Chest Hospital, Shanghai Jiao Tong University, Shanghai 200030, China; yingjiasun@126.com (Y.S.); liziming1980@163.com (Z.L.); jianhong1964@163.com (H.J.); liliangxia@126.com (L.X.)

**Keywords:** non-small cell lung cancer, KRAS mutation, co-mutation, first-line treatment, immune checkpoint inhibitor

## Abstract

(1) Background: The purpose was to systematically assess the impact of KRAS subtypes and co-mutations on responses of first-line treatment and outcomes by genetic classification in advanced KRAS mutant NSCLC. (2) Methods: Molecular pathology was confirmed with NGS; Kaplan–Meier analysis and Cox multivariate model were used to analyze the efficacy of first-line treatment and prognosis in KRAS subgroups. (3) Results: Advanced KRAS mutant NSCLC was confirmed among 183 patients, who received first-line therapy. The most common KRAS subtype and co-mutation were G12C (29.5%) and TP53 (59.6%). ICIs/CHE group prolonged PFS to 16.9 m, vs. (CHE)4.6 m vs. (CHE/BEV)7.0 m (*p* < 0.0001); mOS (ICIs/CHE)37.1 m vs. (CHE)19.8 m vs. [CHE/BEV] 20.7 m (*p* = 0.024). PFS benefited to different degrees after first-line ICI-based treatment in each genetic classification. KRAS G12D even benefited from OS (*p* = 0.045). CHE/BEV prolonged mPFS of KRAS/STK11 co-mutation (*p* = 0.043), but decreased mPFS in G12A subtype (*p* = 0.026). Multivariate analysis indicated that heavy smoking history (≥20 pack-years) (HR = 0.45, *p* = 0.039) predicts optimistic prognosis; PS score 1 (HR = 3.604, *p* = 0.002) and KRAS/SMAD4 co-mutation (HR = 4.293, *p* = 0.027) remained as independent predictors of shorter OS. (4) Conclusions: First-line treatment with ICI benefited KRAS-mutant-NSCLC patients and resulted in non-negative predictive value for any genetic classification. Bevacizumab should be cautiously chosen for patients with KRAS G12A subtype but is recommended for KRAS/STK11 patients. KRAS/SMAD4 is a new co-mutation genotype that displayed independent risk prognostic factors in patients with advanced KRAS-mutant NSCLC.

## 1. Introduction

KRAS mutation is detected in about 30% of cases of NSCLC in an advanced stage in the Western population [1], and the incidence rate in the Chinese population is 9.8% [2], which shows significant ethnic differences. Primary KRAS mutation was usually detected as an exclusive driver mutation, which was mutually exclusive with other driver genes, such as EGFR/ALK/BRAF/ROS1 [3,4,5]. As an essential driving gene, KRAS has been recognized as an undruggable gene in the past few decades. The novel KRAS G12C inhibitors led by Sotorasib [6] being approved by the FDA as second-line treatment made KRAS mutation receive unprecedented attention again.

Most KRAS mutations occur in codon 12, whereas codons 13 and 61 are much less frequently mutated. The most common KRAS mutation is G to T (G > T) transversion: amino-acid replacements are glycine (Gly or G) by cysteine (Cys or C) (G12C) or valine (Val or V) (G12V). G to A (G > A) transitions result in the substitution of glycine for aspartate (Asp or D) (G12D) [7,8]. On the other hand, KRAS mutations often co-occur with non-oncogenes, such as TP53, STK11, and PIK3CA. There is increasing evidence that KRAS-mutant NSCLC does not simply represent a homogeneous subgroup of lung cancer [9,10,11]. Various mutation subtypes and co-mutations of the KRAS gene constitute a complex genotype profile. Each classification shows unique genetic and biological characteristics, which lead to different clinical outcomes of KRAS-mutant NSCLC [1,5,9,12,13,14,15].

Nowadays, the first-line treatment for KRAS-mutant NSCLC is platinum-based chemotherapy combined with or without ICIs [16,17]. However, the results of clinical trials are stratified by KRAS-mutant versus KRAS-wild-type patients [18,19]. The reported studies have not analyzed first-line efficacy by systematically classifying the heterogeneity of KRAS-mutant patients, especially for the Chinese population. Our study screened untreated patients with advanced KRAS-mutant NSCLC. It explored the differences in efficacy in respective first-line treatments by genotyping them through the analysis of genome profiles to excavate the efficacy predictors of KRAS-mutant lung cancer and the prognostic value of different genotypes.

## 2. Materials and Methods

### 2.1. Study Objectives

The primary objective of our study was to describe the progression-free survival (PFS) and overall survival (OS) of NSCLC patients who received first-line treatment in each subgroup carrying KRAS mutation, including subtypes and co-mutations. We also analyzed the outcome of patients according to smoking status, ECOG score, and PD-L1 expression.

### 2.2. Patients

From October 2017 to September 2020, patients were screened in the study if they were diagnosed as having advanced KRAS-mutant NSCLC with a baseline Eastern Cooperative Oncology Group (ECOG) performance-status score of 0 or 1 and receiving standard first-line treatment with complete follow-up information in Shanghai Chest Hospital. Tumor tissues were available for two pathologists evaluated independently, and the PD-L1 expression was displayed as TPS (PD-L1 IHC 22C3 pharmDx kit). Molecular pathology was confirmed with 68 gene panel NGS (Burning Rock Biotech, Guangzhou, China). Patients with other driver genes (EGFR, ALK, ROS1, BRAF, MET, etc.) were excluded.

### 2.3. Study Design and Assessments

Patients were divided into three groups according to different first-line treatments. (1) ICIs/CHE: patients received treatment containing ICIs, combined or not combined with chemotherapy. (2) CHE: patients received platinum-based chemotherapy only. (3) CHE/BEV: patients received platinum-based chemotherapy combined with Bevacizumab. Stratified analysis was performed according to KRAS subtypes, co-mutations, and PD-L1 expression.

### 2.4. End Points and Assessments

The endpoints were PFS and OS. PFS was measured from the date of the first-line treatment to the date of progression or last follow-up. OS was measured from the date of pathological diagnosis of advanced lung cancer to the date of death or last follow-up.

### 2.5. Capture-Based Targeted DNA Sequencing

DNA was extracted with the QIAamp DNA FFPE tissue kit (Qiagen, Hilden, Germany), using Covaris M220 (Covaris, MA, USA) sheared DNA. Fragments of size 200–400 base pairs (bp) were selected with a bead (Agencourt AMPure XP Kit, Beckman Coulter, Brea, CA, USA). DNA template was hybridized with the capture probe bait, and then the hybrid was selected again with magnetic beads and amplified by PCR. Qubit 2.0 fluorometer with the dsDNA high-sensitivity assay kit (Life Technologies, Carlsbad, CA, USA) was used to measure DNA concentration and quality. Samples were sequenced on Nextseq500 sequencer (Illumina, Inc., Madison, WI, USA). The genomic profiles were assessed with 68 gene Lung Core panel from Burning Rock Biotech (Guangzhou, China).

### 2.6. Sequence-Data Analysis

Sequence data were mapped to the reference human genome (hg19) using Burrows-Wheeler aligner (v.0.7.10, Heng Li, Cambridge, UK). Genome Analysis Tool Kit (v.3.2, Aaron McKenna, MA, USA) and VarScan (v.2, Daniel C Koboldt, MO, USA) were used to process local alignment optimization, variant calling, and annotation. Variants with depth less than 100 were filtered using the VarScan. There was a minimum of five supporting reads for INDELs and eight for SNV calling. SNP, which is defined as variants with population frequency over 0.1% in ExAC, 1000 Genomes, dbSNP, ESP6500SI-V2 database, was excluded from the analysis. The remaining variants were annotated with ANNOVAR (v.3, Kai Wang, PA, USA) and SnpEff (v3.6, Pablo Cingolani, MI, USA). DNA translocation analysis was performed using Tophat2 (Daehwan Kim, MD, USA) and Factera (v.1.4.3, Aaron M. Newman, CA, USA).

### 2.7. Statistical Analysis

SPSS (version 25.0, Norman H. Nie, CHI, USA) statistical software was used for data processing and analysis. Bioinformatics analyses were performed with R software (version 3.4.0, Ross Ihaka, Auckland, New Zealand). Pearson chi-square test and Fisher’s exact test were used to compare categorical data. The median follow-up time was calculated with the reverse Kaplan–Meier method. PFS and OS between different groups were compared with a Kaplan–Meier curve. Cox proportional hazard model was used to analyze the prognostic value by removing confounding factors. The results are expressed as hazard ratio (HR), 95% confidence interval, and *p*-value. Statistically significant difference was defined as the *p*-value < 0.05.

## 3. Results

### 3.1. Clinical Characteristics and Genotype Subsection

From October 2017 to September 2020, KRAS-mutant lung cancer was detected, using NGS, in 1132 patients in Shanghai Chest Hospital, Shanghai Jiaotong University. A total of 183 patients who received standard first-line treatment were identified according to the inclusion criteria. A total of 76 cases were screened in group ICIs/CHE, 74 cases in group CHE, and 33 cases in CHE/BEV. The baseline characteristics of patients are shown in Table 1. Of all patients, the average age was 63.1 years. The proportion of male patients was far higher than females (86.9% vs. 17.1%). Heavy smokers were defined as more than 20 pack-years, which accounted for more of the sample (55.7%). The primary pathological type was adenocarcinoma (82.5%), and most of the lesions occurred in the upper lobe lung (70%). Among patients, 80.9% were diagnosed as stage IV at the first visit. Intrathoracic metastasis accounted for 45.4%, and the most distant metastasis organ was bone (43.2%), followed by brain (14.8%). PD-L1 expression was known in 114/183 patients. A total of 80/114 patients (70.2%) showed TPS ≥ 1%, and were more inclined to ICIs/CHE group (*p* = 0.003).

In our study, the most common KRAS-mutation subtype was KRAS G12C (29.5%, 54/183 cases), followed by KRAS G12V (17.5%, 32/183 cases), KRAS G12D (15.3%, 28/183 cases), and KRAS G12A (13.1%, 24/183 cases). In addition, KRAS mutations in codon 13 were 6.6% (12/183 cases), and KRAS codon 61 mutations were 10.4% (19/183 cases, 14 cases of Q61H, 3 cases of Q61L, 1 case of Q61R, and 1 case of Q61K) (Figure 1).

Molecular profiles of KRAS-mutant NSCLC were sequenced with 68 lung cancer-related gene NGS. This showed that TP53 was the most frequent co-mutation in our study population (59.6%, 109/183 cases), followed by STK11 (18.6%, 34/183 cases). In addition, ATM (14.2%, 25/183 cases), PIK3CA (8.7%, 15/183 cases), and SMAD4 (5.5%, 10/183 cases) were common co-mutations. Figure 2 shows the molecular profiles of patients diagnosed with advanced KRAS-mutant NSCLC classified based on gender, stage of disease, KRAS subtypes, smoking status, first-line treatment group, and PD-L1 expression.

### 3.2. Efficacy

#### 3.2.1. Overall Population

The median follow-up time was 23 months, with September 2021 as the cut-off date. 133 patients had a PFS event endpoint, and 84 cases reached an OS endpoint. In the entire cohort, the median PFS was 7.2 months (95% CI 6.0–8.4 m). The median PFS (in months) for each treatment group was 16.9 m (11.1–22.7 m) for ICIs/CHE, 4.6 m (3.4–5.8 m) for CHE, and 7.0 m (5.6–8.4 m) for CHE/BEV (*p* < 0.0001). The stratified risk ratio of disease progression in the ICIs/CHE group was lower than in others (HR (ICIs/CHE vs. CHE] = 0.335, 95% CI: 0.221–0.508, *p* < 0.0001; HR (ICIs/CHE vs. CHE/BEV] = 0.405, 95% CI: 0.249–0.658, *p* < 0.0001) (Figure 3A). As for the cut-off date, the median OS (in months) of the ICIs/CHE group was more prolonged than the CHE group (37.1 m vs. 19.8 m; HR = 0.499, 95% CI: 0.297–0.838, *p* = 0.009) or the CHE/BEV group (37.1 m vs. 20.7 m; HR = 0.49, 95% CI: 0.264–0.91, *p* = 0.024) (Figure 3B). The objective response rate was higher in the ICIs/CHE group (47.4%) than in the CHE group (31.1%) and the CHE/BEV group (21.2%) (Figure 3C).

#### 3.2.2. Value of KRAS Subtypes

PFS in the first-line ICIs/CHE-treatment group showed different degrees of prolongation after stratified analysis of each KRAS subtype (Figure 4). The KRAS G12D subtype can benefit from OS in the ICIs/CHE group (*p* = 0.045). KRAS G12C mutation is the most common subtype in both Chinese and Western populations, and first-line ICI treatment achieved PFS benefits compared with non-ICI groups (ICIs/CHE vs. CHE vs. CHE/BEV = 7.2 m [95% CI: 4.1–10.2] vs. 4 m [95% CI: 3.7–4.2] vs. 5 m [95% CI: 0–10.5], *p* = 0.014). However, compared with non-G12C patients, mPFS in the ICI group was G12C 7.2 m vs. non-G12C 20.0 m (HR (G12C) = 2.424, 95% CI: 1.251–4.696, *p* = 0.009). The same results were found in the CHE group; mPFS was G12C 4.0 m vs. non-G12C 5.6 m (*p* = 0.014). In the KRAS G12A subtype, we found the risk of disease progression was higher in the CHE/BEV group than in the CHE group (HR = 3.742, 95% CI: 0.917–15.273, *p* = 0.066). Analyzing BEV/CHE group data separately showed that mPFS decreased significantly in G12A compared with non-G12A patients (3.2 m vs.7.7 m, *p* = 0.026).

#### 3.2.3. Value of Co-Occurring Mutations

Co-occurring mutation, as an important stratification factor of KRAS-mutant NSCLC, was analyzed. The analysis showed that KRAS/TP53 significantly prolonged mPFS to 18.7 m in the first-line ICIs/CHE group versus 6.1 m in the CHE group and 6.8 m in the CHE/BEV group (*p* < 0.0001). For patients with KRAS/ATM co-mutation, chemotherapy alone will lead to PFS shortening (*p* = 0.015). Results showed a tendency to benefit from mPFS in the CHE/BEV group for patients who carried KRAS/STK11 co-mutation, although the value was in a critical state (ICIs/CHE vs. CHE vs. CHE/BEV = 4.4 m [95% CI: 2.6–6.1 m] vs. 3.9 m [95% CI: 3.0–4.7 m] vs. 7.0 m [95% CI: 5.0–9.0 m], *p* = 0.043). For patients with KRAS/SMAD4, there was no difference in the efficacy of the three treatment methods (*p* = 0.699) (Figure 5). In patients with STK11 co-mutation, we additionally found the efficacy in the ICIs/CHE group was significantly worse than non-STK11 (mPFS (STK11) 4.4 m vs. (non-STK11) 19.1 m, HR (STK11) = 2.808, 95% CI: 1.255–6.283, *p* = 0.012); the same in CHE group, mPFS = (STK11) 3.9 m vs. (non-STK11) 5.6 m (*p* = 0.004).

### 3.3. Prognosis

Univariate analysis based on each KRAS subtype showed the prognosis of G12V subtype was significantly better than non-G12V; mOS was 39.7 m vs. 21.3 m (HR(G12V) = 0.373, 95% CI: 0.179–0.777, *p* = 0.008). However, Q61x was a poor prognostic factor in which mOS was only 13.4 m vs. 29.7 m in non-Q61x (HR(Q61x) = 2.017, 95% CI: 1.129–3.604, *p* = 0.028). In addition, the G12C subtype showed a poor prognostic trend without statistical significance (HR(G12C) = 1.424, 95% CI: 0.901–2.248, *p* = 0.13) (Figure 6A).

As for co-mutation, the results showed that KRAS/SMAD4 type was significantly correlated with poor prognosis. Compared with the non-SMAD4 group, mOS was 8.5 months in KRAS/SMAD4 vs. 24.4 months in KRAS/non-SMAD4 (HR(SMAD4) = 2.59, 95% CI: 1.187–5.651, *p* = 0.035). STK11 also predicted a poor prognosis, although it did not reach statistical difference, but increased the risk of death by 45% (HR(STK11) = 1.445, 95% CI: 0.866–2.413, *p* = 0.159) (Figure 6B).

Clinical and molecular variables were included in Cox regression analysis. The univariate analysis identified gender, smoker, PS score, stage of disease, KRAS mutation G12V or Q61x, and SMAD4 co-mutation as factors affecting the OS. In addition, the expression of PD-L1 (stratified by TPS ≥ 1%), STK11, gender, and smoking history were added as factors that may affect the outcome of Cox analysis (Table 2). In the multivariate Cox model, KRAS/SMAD (HR = 4.239, *p* = 0.027) and PS1′ (HR = 3.604, *p* = 0.002) remained as independent factors for poor prognosis. In contrast, the history of heavy smoking showed independent good prognostic value (HR = 0.45, *p* = 0.039) (Table 2).

## 4. Discussion

With the development of treatment and the popularization of NGS, a first-line treatment scheme for NSCLC patients is to bring forth the new through the old. KRAS G12C inhibitors, such as AMG510 (Sotorasib) and MRTX849 (Adagrasib), have only been approved by FDA for second-line treatment. For a long time, due to the unique biological heterogeneity and ethnic differences of KRAS mutation, the reported research conclusions have been controversial. We report the first intergroup study of KRAS patients in the Asian race with a complete follow-up process from the initial diagnosis stage, showing the difference in therapeutic effect and prognosis of advanced KRAS-mutant NSCLC in three first-line treatments under stratified factors according to molecular profiles.

The genomic profile directly reflects the differences between Chinese and Western populations. KRAS is the first driving gene in Western populations, and the proportion of mutant subtypes is G12C (46%) and G12V (16%) [12]. On the other hand, in the Chinese population, KRAS incidence is lower than EGFR. In our cohort, the subtype distribution is G12C (29.5%), G12V (17.5%), and G12D (15.3%), which is consistent with the epidemiological survey in China [20,21]. In terms of CO mutation, our study showed that TP53 (59.6%) is significantly higher than 42% of the Western population. The incidence of STK11 in 18.6% is lower than that in the Western population (29%) [9].

Nowadays, first-line standard treatment of KRAS-mutant lung cancer is based on chemotherapy, combined with or without ICIs [16,22]. Our study of the efficacy of KRAS mutation in the overall population concluded that ICIs/CHE significantly prolonged PFS (*p* < 0.0001) and OS (*p* = 0.014), which is consistent with the existing reported conclusions [18,19]. KRAS mutations are more often associated with heavy smoking, higher tumor mutation burden (TMB), and PD-L1 expression mediated by ERK activation and a high level of tumor-infiltrating lymphocytes (TILs) [23,24]. By analyzing KRAS subtypes, it was found that the PFS of each subtype in the first-line ICI/Che group was prolonged to varying degrees. Nevertheless, only KRAS G12D type had survival benefits (*p* = 0.045). This critical value result may be related to the independent adverse prognostic value of KRAS G12D [25]. The potential beneficial effect of ICIs in KRAS G12D patients suggested that immunotherapy should be the recommended treatment for the G12D subtype. KRAS/TP53 in the co-mutant sort indicates high expression of immune checkpoint effector molecules [23,26]. In our cohort, it was observed that KRAS/TP53 obtained a significant prolongation of PFS in the ICIs/CHE group (*p* < 0.0001). STK11 is an adverse prognostic factor due to its inert immunosuppressive environment [27]. We found no significant difference between first-line ICI treatment versus other regimens in KRAS/STK11 patients (*p* = 0.126), even if it could reduce the risk of disease progression by 52%, suggesting that KRAS/STK11 is not a contraindication of immunotherapy.

Few reports on the effect of antiangiogenic therapy on KRAS-mutant lung cancer indicated that, compared with chemotherapy, the addition of Bevacizumab seemed to have an adverse treatment response [28,29]. In our study, hierarchical analysis suggested that the addition of Bevacizumab based on chemotherapy cannot improve the PFS or OS. We even found that the KRAS G12A subtype was an adverse predictor of Bevacizumab (it increased the risk of disease progression by 3.7 times compared with chemotherapy only, *p* = 0.026). We observed that chemotherapy plus Bevacizumab benefited the PFS of KRAS/STK11 co-mutation type (*p* = 0.043). Bonanno et al. found in the PDX model that Bevacizumab reduced the risk of death in NSCLC patients with STK11 mutation by downregulating the activation of AMPK [30]. Analysis of IMPOWER 150 also presented that, for the subgroup of patients with KRAS-mutant lung cancer carrying STK11 and/or Keap1 simultaneously, Atezolizumab combined with Bevacizumab and chemotherapy is an effective first-line treatment [5]. These results suggest that antiangiogenic therapy may be a treatment scheme in KRAS/STK11. Look forward to future research to confirm these concepts. Based on the above, our results showed the different impacts of KRAS-mutation subtypes and co-mutations on the efficacy of first-line treatment. We propose that the treatment strategies for KRAS-mutant patients should be based on the status of subtypes and co-mutations.

The prognostic value of KRAS-mutant NSCLC has been controversial since the 1980s [13,31], with researchers revealing that the prognosis was worse for patients in China [32]. Although smoking is a clear risk factor for inducing cancer, our results observed that the OS of heavy smokers was prolonged after Cox model correction. Meta-analysis suggested that smokers may benefit more from immunotherapy due to the influence of cigarettes on the tumor microenvironment [33]. The positive prognosis in our study of heavy smokers may be related to the fact that 41.5% of cases received ICIs-included treatment. Our results showed that KRAS mutation subtype did not have an independent prognostic value in multivariate correction. Univariate conclusion: the benefit of OS in patients with the KRAS G12V subtype may be due to the positive efficacy of ICIs by G12V mutation upregulating PD-L1 expression through the TGF-β/EMT-signaling pathway [34]. Western and Chinese studies suggested that the KRAS G12C subtype had a worse prognosis than KRAS wild-type NSCLC [2,12]. In our cohort, the G12C subtype was observed with an increased risk of death of 42.4% versus non-G12C patients. It suggested that even compared with KRAS non-G12C patients, the KRAS G12C subtype also manifested a potential negative prognosis.

There is growing evidence that the presence of co-mutations affects tumor characteristics and biological behavior, which alter prognosis [5]. However, there are inconsistent conclusions concerning the significance of prognosis [9,35]. The analysis result of our study suggested that the most co-mutation type KRAS/TP53 did not have prognostic value, and although KRAS/STK11 shows a downward trend in OS, the statistical difference in prognostic value is not significant. SMAD4 co-mutation was an independent negative prognostic value in Kaplan–Meier univariate analysis and COX multivariate adjustment. Therefore, we proposed KRAS/SMAD4 as a novel classification of KRAS mutant NSCLC, which manifested as a low rate of response to standard first-line treatment. SMAD4 plays the critical intracellular signal transduction mediator in the TGF-β pathway; the protein deletion is currently associated with lymph-node metastases, increased angiogenesis in vitro, and more aggressive tumor behavior in patients [36,37]. Our study showed that the incidence of SMAD4 co-mutations in KRAS-mutant lung-cancer patients was 5.5%, higher than in the NSCLC population at 2.5%. Furthermore, the KRAS/SMAD4 co-mutant type seemed to be associated with low expression of PD-L1. Research demonstrated that SMAD4-mediated binding to the Rorc locus promotes the differentiation and maturation of Th17 lymphocytes [38], which exacerbate cancer migration and metastatic spreading [39]. The role of KRAS/SMAD4 in the tumor microenvironment has additional value for exploration.

There may be some survivor bias in our data due to the retrospective study, and some classifications involved a small sample size. The results can be further supported by expanding the sample size and conducting prospective studies. The conclusions of our study are intended to provide clinicians with a reference for making first-line therapeutic regimes for patients with KRAS-mutant NSCLC.

## 5. Conclusions

In conclusion, the clinical outcome of KRAS-mutant NSCLC patients benefits from a first-line ICI-based therapeutic regime to varying degrees. Outcomes for patients with KRAS/SMAD4 co-mutation were inferior, which we propose as a novel and treatment-insensitive genotype. Patients with KRAS G12A subtype should avoid Bevacizumab; however, it may become a treatment option for KRAS/STK11-co-mutation patients.

## Figures and Tables

**Figure 1 jcm-11-04003-f001:**
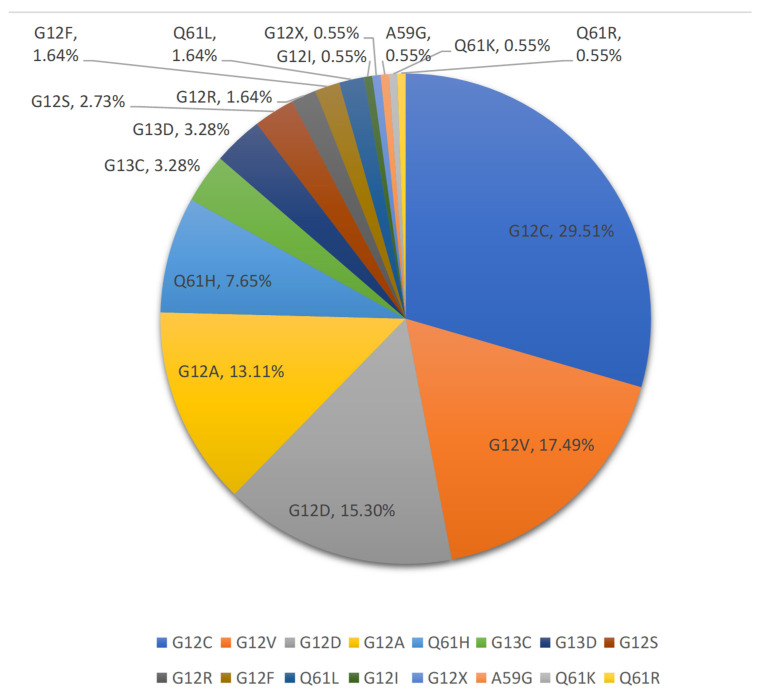
The proportion of KRAS-mutation subtypes.

**Figure 2 jcm-11-04003-f002:**
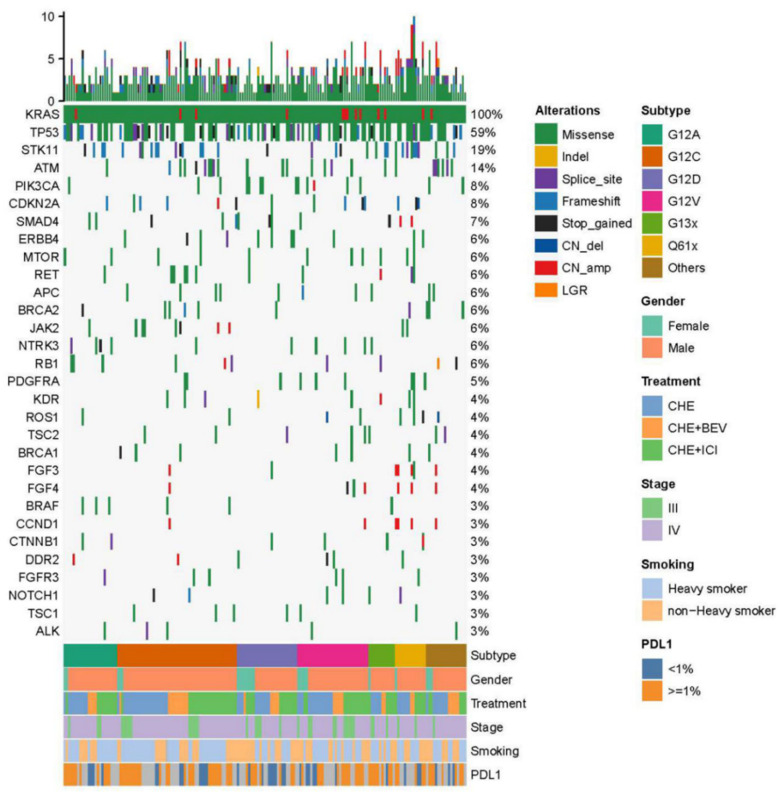
Molecular profile of KRAS-mutant NSCLC (non-small cell lung cancer, NSCLC); Stratification by KRAS subtype, gender, first-line treatment, stage of disease, smoking status, and PD-L1 expression.

**Figure 3 jcm-11-04003-f003:**
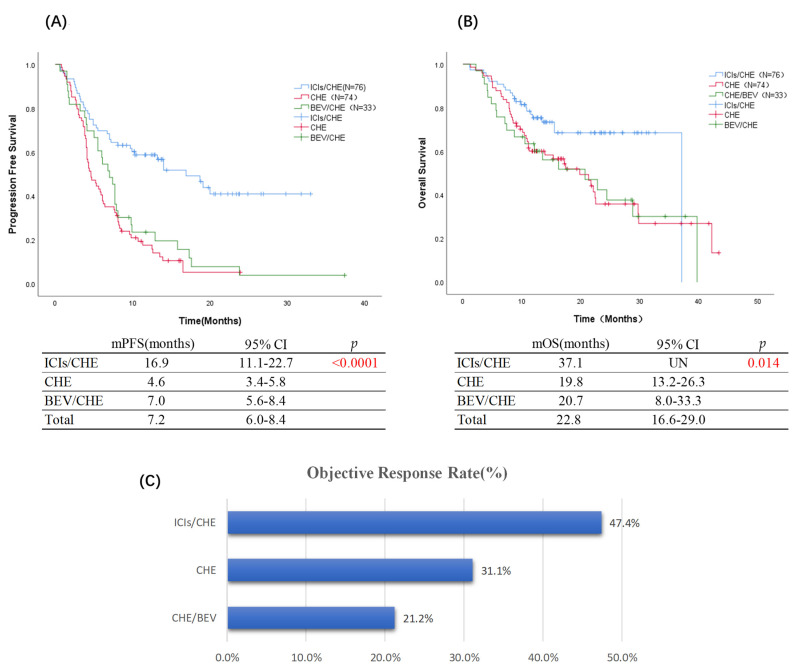
KaplanMeier estimates in the whole cohort (*p* < 0.05, statistical significance; *p* < 0.001, significant statistical significance, red indicates); (**A**) Progression-free survival (ICIs/CHE vs. CHE vs. CHE/BEV = 16.9 months vs. 4.6 months vs. 7.0 months, *p* < 0.0001); (**B**) Overall survival (ICIs/CHE vs. CHE vs. CHE/BEV = 37.1 months vs. 19.8 months vs. 20.7 months, *p* = 0.014); (**C**) Objective Response Rate (ICIs/CHE vs. CHE vs. CHE/BEV = 47.4% vs. 31.1% vs. 21.2%).

**Figure 4 jcm-11-04003-f004:**
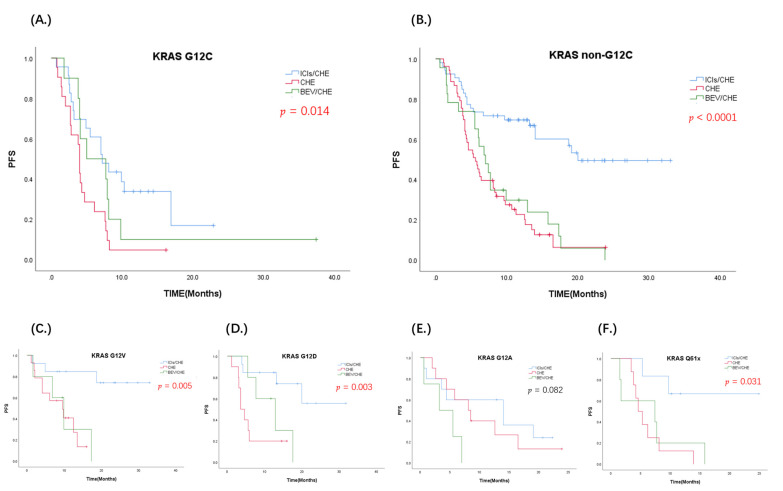
Kaplan–Meier PFS curve of three treatment groups in each KRAS subtype; The blue curve represents ICIs/CHE group, the red curve represents CHE group, and the green line represents CHE/BEV group (*p* < 0.05, statistical significance; *p* < 0.001, significant statistical significance, red indicates. *p* ≥ 0.05, no statistical significance, black indicates); (**A**) KRAS G12C; (**B**) KRAS non-G12C; (**C**) KRAS G12V; (**D**) KRAS G12D; (**E**) KRAS G12A; (**F**) KRAS Q61x.

**Figure 5 jcm-11-04003-f005:**
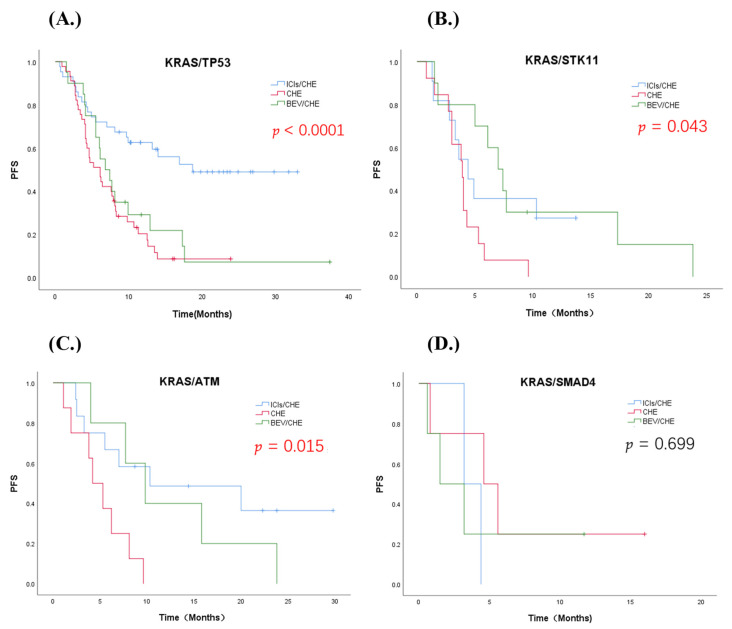
Kaplan–Meier PFS curve of three treatment groups in KRAS-co-mutation classification (*p* < 0.05, statistical significance; *p* < 0.001, significant statistical significance, red indicates. *p* ≥ 0.05, no statistical significance, black indicates); (**A**) KRAS/TP53; (**B**) KRAS/STK11; (**C**) KRAS/ATM; (**D**) KRAS/SMAD4.

**Figure 6 jcm-11-04003-f006:**
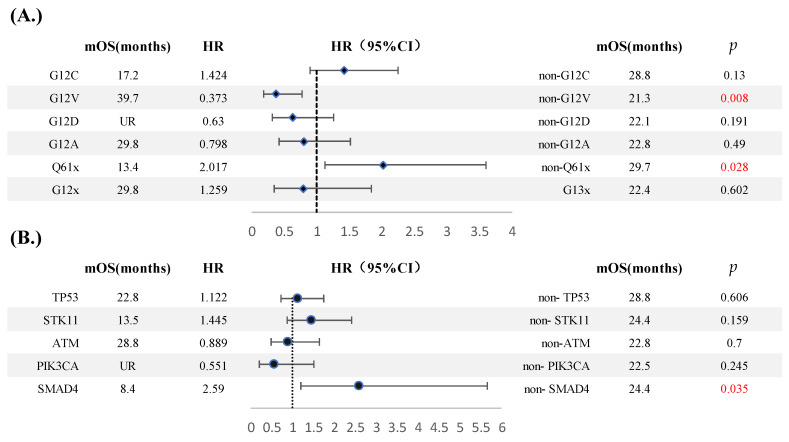
Forest plot of outcomes in KRAS subtypes (**A**) and co-mutations (**B**). (*p* < 0.05, statistical significance; *p* < 0.001, significant statistical significance, red indicates. *p* ≥ 0.05, no statistical significance, black indicates).

**Table 1 jcm-11-04003-t001:** Baseline patient characteristics.

	CHE/ICIs (N = 76)	CHE (N = 74)	CHE/BEV (N = 33)	Total (N = 183)	*p*-Value
**Age—years old**					
Median	62.1	64.1	63.1	63.1	
Range	31–81	47–76	47–77	31–81	
**Sex (%)**					0.278
Male	63 (82.9%)	65 (87.8%)	31 (93.9%)	159 (86.9%)	
Female	13 (17.1%)	9 (12.2%)	2 (6.1%)	24 (13.1%)	
**Disease stage (%)**					0.094
III	18 (23.7%)	15 (20.3%)	2 (6.1%)	35 (19.1%)	
IV	58 (76.3%)	59 (79.7%)	31 (93.9%)	148 (80.9%)	
**Smoking status (%)**					0.569
More than 20 pack-years	43 (56.6%)	44 (59.5%)	16 (48.5%)	100 (54.6%)	
Less than 20 pack-years	33 (43.4%)	30 (40.5%)	17 (51.5%)	83 (45.4%)	
**Lesion location**					0.630
Upper left	20 (26.3%)	25 (33.8%)	13 (39.4%)	58 (31.7%)	
Lower left	9 (11.8%)	10 (13.5%)	2 (6.1%)	21 (11.5%)	
Upper right	32 (42.1%)	26 (35.1%)	12 (36.4%)	70 (38.3%)	
Right middle	5 (6.6%)	1 (1.4%)	1 (3.0%)	7 (3.8%)	
Lower right	10 (13.2%)	12 (16.2%)	4 (12.1%)	26 (14.2%)	
**ECOG PS score (%)**					0.209
0	33 (%)	24 (32.4%)	16 (48.5%)	73 (39.9%)	
1	43 (%)	50 (67.6%)	17 (51.5%)	110 (60.1%)	
**Histologic type (%)**					0.174
Adenocarcinoma	59 (77.6%)	61 (82.4%)	31 (93.9%)	151 (82.5%)	
Squamous carcinoma	3 (3.9%)	3 (4.1%)	0 (0%)	6 (3.3%)	
NSCLC not otherwise specified (NSCLC-NOS)	14 (18.4%)	8 (10.8%)	2 (6.1%)	22 (12.0%)	
Neuroendocrine carcinoma	0 (0%)	2 (2.7%)	0 (0%)	2 (1.1%)	
**Metastatic organ (%)**					0.847
Intrathoracic	25 (32.9%)	37 (50.0%)	21 (63.6%)	83 (45.4%)	
Brain	10 (13.2%)	8 (10.8%)	9 (27.3%)	27 (14.8%)	
Bone	28 (36.8%)	35 (47.3%)	16 (48.5%)	79 (43.2%)	
Adrenal gland	11 (14.5%)	6 (8.1%)	5 (15.2%)	22 (12.0%)	
Liver	3 (4.0%)	3 (4.1%)	2 (6.1%)	8 (4.4%)	
Others	9 (11.8%)	9 (12.2%)	5 (15.2%)	23 (12.6%)	
**PD-L1 expression (%)**	N = 56/76(73.7%)	N = 40/74(54.1%)	N = 18/33 (54.5%)	N = 114/183 (62.3%)	0.003
0	12 (21.4%)	11(27.5%)	11 (61.1%)	34 (29.8%)	
≥1%	44 (78.6%)	29(72.5%)	7 (38.9%)	80 (70.2%)	

**Table 2 jcm-11-04003-t002:** Prognostic correlation factors (* *p* < 0.05, ** *p* < 0.001).

Variable	Univariate Analysis			Multivariate Analysis		
	HR	95% CI	*p*	HR	95% CI	*p*
**Gender**						
Male	1					
Female	0.593	0.273–1.287	0.186	0.481	0.172–1.347	0.164
**performance status, PS**						
0	1					
1	2.212	1.379–3.546	0.001 **	3.604	1.630–7.967	0.002 *
**Stage**						
III	1					
IV	2.722	1.255–5.908	0.011 *	1.191	0.473–2.999	0.711
**Smoker**						
<20 pack-year	1					
≥20 pack-year	0.958	0.620–1.481	0.847	0.450	0.210–0.962	0.039 *
**PD-L1**						
0%	1					
≥1%	0.292	0.377–1.341	0.711	1.504	0.629–3.595	0.359
**G12V**						
non-G12V	1					
G12V	0.373	0.179–0.777	0.008 *	0.432	0.148–1.257	0.124
**Q61x**						
non-Q61x	1					
Q61x	1.961	1.079–3.563	0.027 *	1.267	0.478–3.358	0.634
**STK11**						
non-STK11	1					
STK11	1.445	1.445–2.413	0.159	2.359	0.947–5.877	0.065
**SMAD4**						
non-SMAD4	1					
SMAD4	2.590	1.187–5.651	0.017 *	4.293	1.179–15.628	0.027 *

## Data Availability

The datasets developed and analyzed during this study are available from the corresponding author upon reasonable request.

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
