# Peer review of "Impact of KRAS Mutation Subtypes and Co-Occurring Mutations on Response and Outcome in Advanced NSCLC Patients following First-Line Treatment"

_jcm, 2022, doi:10.3390/jcm11144003_

Round 1

Reviewer 1 Report

Thank you for your submission, I think the literature review of KRAS gene subtypes and co-mutation and the effect of chemotherapy on each is very important data in the current treatment of lung cancer with the advent of molecular targeted drugs. The number of cases is not small, and I read the article with great interest. As far as I can tell from a simple search, there seems to be a paper showing that KRAS/SMAD4 significantly worsens prognosis in pancreatic adenocarcinoma, which seems to be consistent with the results (Yokose T, et al. Cancer Sci 2020.).

I would like to make a few points and ask a few questions.

1. In the Abstract, there is no mention of "(1)". (2) (3) and after are listed, so it is probably an omission.

2. In line 264 of the Discussion, the word "Atelizumab" is mentioned, but is it a mistake for "Atezolizumab"?

3. Were there any cases that included both I-O and Bev, such as the IMpower150 regimen? If included, which group did you classify them in? As you stated, a subgroup analysis of the IMpower150 trial showed that these combinations may have favorable outcomes, but how about this study?

4. I believe that it is the role of the oncologist to promote smoking cessation. with that in mind, do you have any discussion regarding the better prognosis in the heavy smokers?

Reviewer 2 Report

Initial treatment data for KRAS-mutated lung cancer is a clinical question with very high needs, and I believe that this paper is meaningful.

However, I will point out some points that I was interested in.

Not bad, but requires some modifications

Minor revisions because I think it's okay to post it if possible

1.

In line 275, ”In our cohort G12C subtype was observed with an increased risk of death of 42.4% versus non-G12C patients. The potential negative prognostic propensity of KRAS G12C subtype is manifested in patients with KRAS mutant lung cancer, not only in NSCLC population. “, But I don't understand the meaning. Is it means “Non-adeno”? You should correct the description.

 ï¼’.

In line 243, it is reported that only G12D prolongs overall survival, but is there any consideration as to the reason for this?

3.

Figure 4 is very impressive and easy to understand, but I feel that the comparison between G12C and non-G12C, which is described in the text is clinically more important. Why don’t you add the KM of G12C vs. non-G12C in figure 4, or replace the KM of KRAS mutational subtype in Figure 4 to supplemental figure?

4.

The effects of the KRAS mutational subtype and the co-mutation subtype on clinical efficacy are well investigated. However, with this result, I would like the authors to indicate whether or not they think that subtype-based treatment strategies should be considered in these groups. In addition, there is currently no change in treatment options other than KRAS G12C. Based on this result, I would like you to consider whether it is better to investigate the KRAS subtype other than G12C, or whether it is only necessary to know whether there is G12C, or whether it is better to investigate co-mutation.
